# The Influence of Freezing and Thawing on the Yield and Energy Consumption of the Celeriac Juice Pressing Process

**Rafał Nadulski [1],\*** , **Zbigniew Kobus [2]** and **Tomasz Guz [1]**

[1]  Department of Food Engineering and Machines, University of Life Sciences in Lublin, 20-950 Lublin, Poland; tomasz.guz@up.lublin.pl

[2]  Department of Technology Fundamentals, University of Life Sciences in Lublin, 20-950 Lublin, Poland; zbigniew.kobus@up.lublin.pl

\*   Correspondence: rafal.nadulski@up.lublin.pl

**Abstract:** The aim of this study is to indicate the influence of pretreatment, consisting of grinding vegetables and then freezing and thawing the raw material before pressing on the process efficiency and quality of obtained juice. The subject of the research was celeriac root (*Apium graveolens* L) of the Edward variety. Juice pressing was carried out in a laboratory basket press. The pressing yield and specific energy were values characterizing the pressing process, while pH and the extracted content in the juice were used to assess the juice quality. The experiment was performed according to three procedures. In each of them, the ground celeriac root (pulp or chips) was initially pretreated through freezing and thawing and then pressed twice. Among the examined methods of obtaining juice, the most beneficial method was pressing juice from the pulp, then freezing and thawing the pomace obtained in the first cycle, and finally, pressing the pomace. It is an energy-optimal method and guarantees a high pressing yield as well as obtaining juice with a higher soluble solid content than during the process of pressing chips.

**Keywords:** celery root; freezing; thawing; pressing; yield; specific energy; juice properties

## 1. Introduction

Vegetables are a crucial element of a healthy diet. Celeriac (*Apium graveolens* L) is a common vegetable, occurring in temperate climate zones, and it has a significant culinary meaning [1]. Celeriac roots, as well as leaves, are edible and can be consumed fresh or processed, e.g., frozen, canned, or as juices. Their high nutritional value arises from their low caloric value, high content of vitamins (C, $B_1$, $B_2$, $B_3$, $B_6$, $B_{12}$, A, D, E), macroelements (Ca, K, Mg, P, S), microelements (Fe, Cu, Zn, Mn, B), and other biologically active elements [2–5]. Celeriac has the following properties: anti-inflammatory, antiallergic, antioxidant, antimicrobial, and antiviral [6–8]. It should also be noted that allergens may also be present in celery juice [9]. There is a large amount of data in the literature regarding the results of examined celeriac roots' chemical properties and their influence on health. However, there is almost no data presenting celeriac juice pressing technologies. Pressing is one of the common methods of acquiring juice from raw materials of plant origin. During the pressing process, the three-phase system (pulp) is separated into liquid (cell juice), gas (air), and solid (pectin, cellulose) states. The efficiency of juice pressing is influenced by operational and construction factors, as well as the properties of the pressed raw material [10–12]. In industrial conditions, basket presses with a drainage hose system or roll and belt presses are most commonly used for pressing. The method of pulp preparation and its pretreatment has a strong impact on the juice quality, as well as the pressing efficiency and energy

consumption [10–13]. The aim of the pretreatment is to destroy cellular structures and then reduce the flow resistance of the liquid through the filling. For the pretreatment of pulp before pressing, the most commonly used method is enzymatic treatment, carried out at elevated temperatures [14,15]. According to Nowak [16], in the case of enzymatic treatment of vegetable pulp, it is recommended to heat the pulp to 110–120 °C, cool it to a temperature of approximately 50 °C, and then enzymatically liquefy it. Research conducted by Wightman and Wrolstad [17,18] indicates that enzymatic treatment may have a negative impact on the pro-health properties of the juice. Hence, new alternative methods of pulp pretreatment are being explored; these methods are based on physical processes. Applying physical methods of pulp treatment may reduce the negative alterations in the raw material and, as a result, in juice [19]. For physical pulp pretreatment, the following methods are used: ohmic heating [20], pulsed electric field [21,22], electro-osmosis [23], radiation [24], microwave heating [25,26], sonication [27], as well as freezing and thawing of the pulp [13,28,29]. Freezing is used to preserve foods. However, for some foods, freezing can be used as a pretreatment to enhance the efficiency of the main processing procedures [30,31]. The application of a freezing and thawing method as a pretreatment process results in the destruction of plant material structures and therefore enables an easier release of cellular juice. With reference to many studies, the freezing process negatively affects the pro-health properties of food only to a small extent, and for that reason, this method is commonly used in the food processing industry [32,33]. Nadulski et al.'s [13,28,29,34] study proves the possibility of maintaining the pro-health properties of fruit and vegetable juices while applying freezing and thawing methods as pulp pretreatment. To the best of our knowledge, there is no evidence regarding the use of freezing and thawing as a method supporting the process of pressing juice from celeriac roots. Scientists [35] have demonstrated that the use of current vegetable processing methods applied in the food industry may have a negative impact on the nutritional properties of the obtained products. Thus, it is essential to develop such a method and then apply such technologies that affect the pro-health properties of used raw materials to a small extent.

The aim of this study is to indicate the influence of pretreatment, consisting of grinding vegetables and then freezing and thawing the raw material before pressing on the process efficiency and quality of the obtained juice. The scope of this research included the preparation of the raw material in accordance with the procedures, determining the impact of the pulp pretreatment on the pressing efficiency through calculating the pressing yield and specific energy, as well as the assessment of selected quality traits of the obtained juice through the determination of the soluble solid content and pH.

## 2. Materials and Methods

### 2.1. Material

The subject of the research was celeriac root (*Apium graveolens* L) of the Edward variety. The raw material was obtained from a specialist horticultural farm (Producer Group "Krasoń"), and after harvesting, it was stored under cooling conditions at a temperature of approximately 10 °C. The moisture of the studied celeriac roots was 86.4%.

### 2.2. Preparation of the Raw Material for Pressing

The research was performed on healthy roots that were not mechanically damaged. The roots were washed in cold water, drained with the use of blotting paper, and eventually ground. The process was conducted with the use of an MKJ250 (Spomasz, Nakło, Poland) shredding machine equipped with two grinding discs that convert the material into pulp or chips [13]. Samples of ground material of 150 g weight were placed in plastic containers. The material was separated into two parts: the first part of the ground material (pulp or chips) was pressed. The second part was initially frozen with the use of a freezer (F6243W, Gorenje, Slovenia) at a temperature of −20 ± 1 °C, and then it was

defrosted in the thawing machine (SUP-4, WAMED, Warsaw, Poland) at a temperature of 22 ± 1 °C and eventually pressed.

## 2.3. Pressing

Juice pressing was carried out in a self-designed laboratory basket press of stainless-steel compatible with an Inston 4302 machine (INSTRON, Norwood, MA, USA). The press consisted of a base, a cylinder with a replaceable perforated bottom with a perforation diameter of 3 mm, and a piston (Figure 1).

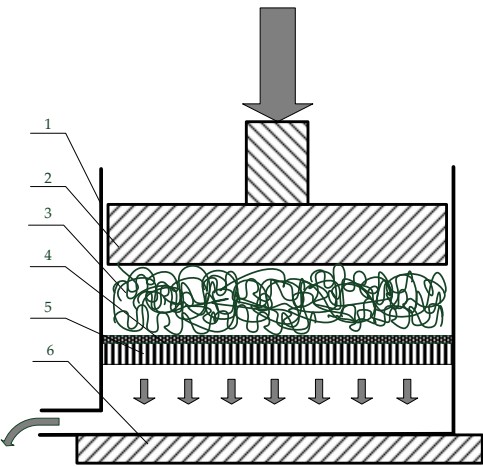

**Figure 1.** Schematic of the press: cylinder, 2—piston, 3—pulp, 4—mesh, 5—perforated bottom, 6—base.

Before pressing the pulp on the surface of the perforated bottom, a mesh with an aperture size of 0.1 × 0.1 mm was introduced [13]. A sample of ground material was placed in the press cylinder and then loaded with a piston. The pressing process continued to the point when the force amounted to 9.5 kN and then stopped. The used pressure of 9.5 kN allowed an average pressure of 1.7 MPa to be reached. The speed of piston displacement was 1 mm s$^{-1}$. Each measure was performed 10 times. The process of the pressing samples according to the selected procedures was performed in two cycles. In the first cycle, pulp or chips were pressed; in the second cycle, the pomace was pressed.

The pressing yield was calculated according to the following formula:

$$Y\ (\%) = M_j/M_i \times 100, \tag{1}$$

where $Y$—pressing yield, %; $M_j$—juice mass obtained by pressing, kg; $M_i$—mass of input material (pulp, chips, or pomace), kg.

The value of press work for each measurement was calculated using the Instron 4302 software (Series IX v. 7.43, Instron, Norwood, MA, USA, 1995). The specific energy of the pressing process was calculated according to the following formula:

$$E\ (\text{kJ·kg}^{-1}) = W/M_i, \tag{2}$$

where $E$—specific energy, kJ·kg$^{-1}$; $W$—press work, kJ; $M_i$—juice mass obtained by pressing, kg.

## 2.4. Pressing Procedures

The experiment was performed according to three procedures depicted in Figure 2. In each procedure, the sample was pressed twice. In procedure A, the initial processing before the first pressing involved grinding. In procedure B, the sample was ground before the first pressing process; the pomace was frozen at a temperature of −20 ± 1 °C, and then after thawing at a temperature of 22 ± 1 °C, it was pressed. In procedure C, the process of freezing at a temperature of −20 ± 1 °C and thawing at a

temperature of 22 ± 1 °C was conducted before both first pressing and second pressing (pomace). The procedure was tested in duplicate.

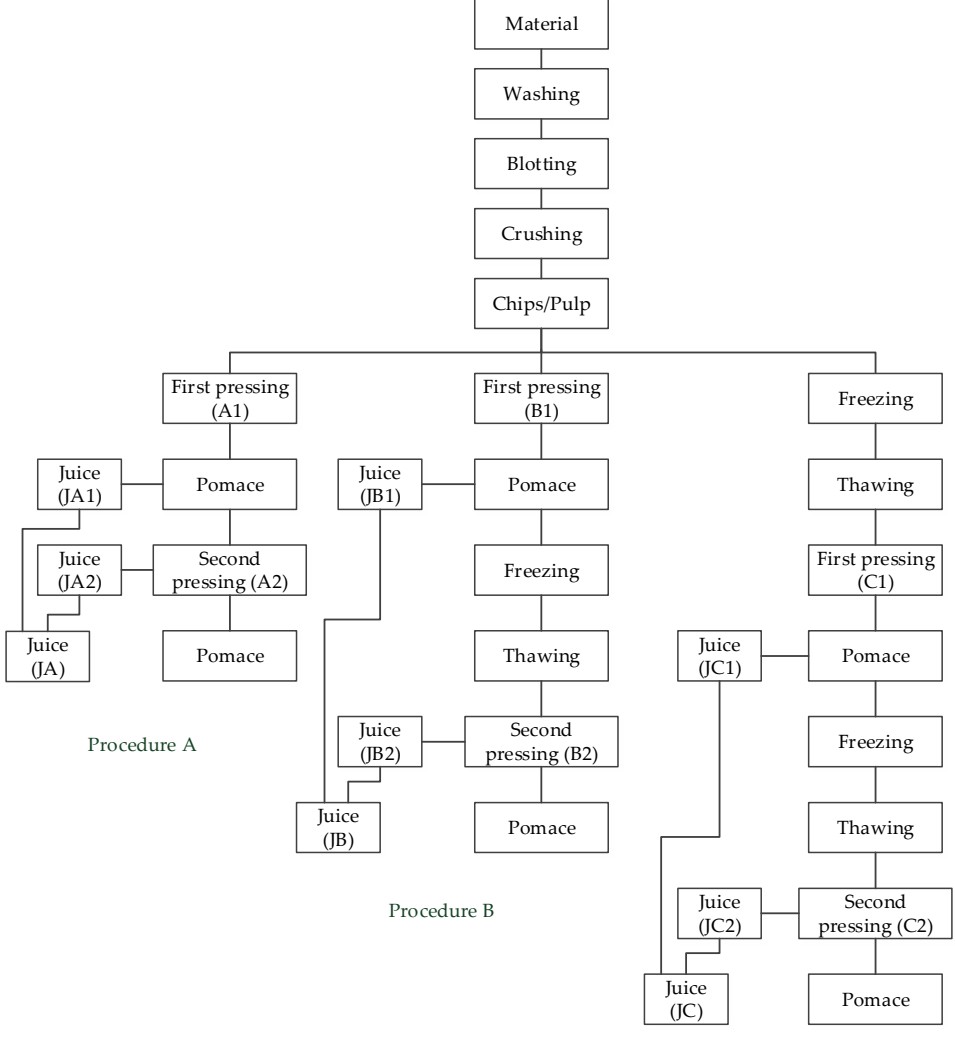

**Figure 2.** Experiment scheme: JA1—juice obtained in procedure A after first pressing; JA2—juice obtained in procedure A after second pressing; JA—juice obtained in procedure A; JB1—juice obtained in procedure B after first pressing; JB2—juice obtained in procedure B after second pressing; JB—juice obtained in procedure B; JC1—juice obtained in procedure C after first pressing; JC2—juice obtained in procedure C after second pressing; JC—juice obtained in procedure C.

## 2.5. Properties of the Juice

After each pressing process, the juice soluble solid content [36] and its pH [37] were determined. Refractometer PAL-1 (Atago, Tokyo, Japan) was used as an instrument to determine the juice extract content, whereas the pH meter CP-411 (Elmetron, Zabrze, Poland) was used to measure the juice pH. Each sample was tested ten times.

## 2.6. Statistical Methods for Analysis

A statistical compilation of the research results was conducted with the use of the ANOVA analysis of variance for factorial designs, which allows the analysis of designs with interactions for selected quality predictors. The pressing yield and specific energy were assumed as the values characterizing the pressing process, whereas the extracted content (°Bx) and pH were assumed as values for assessing

the juice quality. The Fisher (LSD-last significant differences) test was used to analyze the significance of differences (Statistica v. 12.0, StatSoft Inc., Tulsa, OK, USA, 2015).

## 3. Results and Discussion

### 3.1. Yield and Specific Energy of Juice Pressing

In the experiment, the total yield and specific energy of pressing juice from ground celeriac root as well as the yield and specific energy of pressing in the first and second cycles were determined for individual research procedures. The statistical analysis proved a significant impact of the chosen procedure and the method of sample grinding before pressing on yield and the specific energy of juice pressing. The total yield, i.e., including two pressing cycles, was in a range from 28.1% to 74.5%, while the specific energy was in a range from 0.51 kJ to 4.22 kJ·kg$^{-1}$. Figure 3 presents the influence of the applied procedures on the total yield $Y_t$ of pressing juice from the celeriac root that was crushed into pulp and chips.

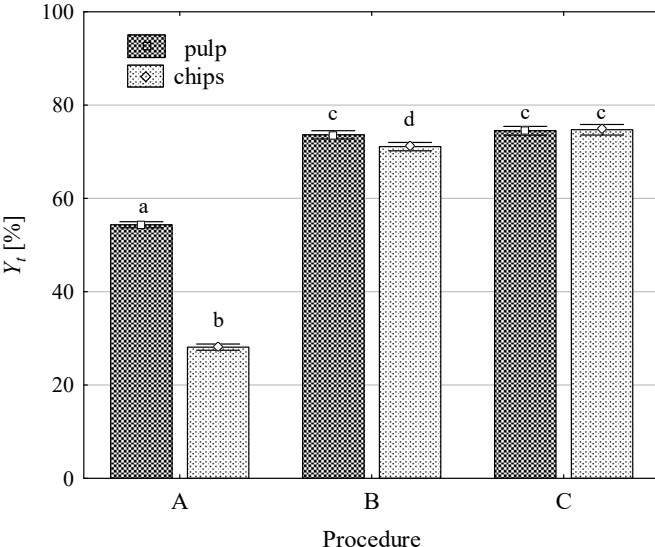

**Figure 3.** Impact of applied procedures on the total yield $Y_t$ of pressing juice from ground celeriac root. A—two-stage pressing of pulp; B—two-stage pressing, first pulp, and then pomace after freezing and thawing; C—two-stage pressing, first pulp after freezing and thawing and then pomace after freezing and thawing. a, b, c, and d—average values marked with the same letter show no statistically significant difference ($p \leq 0.05$).

For procedure A, the total pressing yield $Y_t$ from celery root pulp amounts to 54.4%, which is 93.4% higher compared with the chips. Sharma [38,39] et al. point out that the pressing yield of root vegetables depends primarily on the degree of material grinding, its temperature, and enzymatic treatment. According to Nadulski et al. [13,28,29], the process of freezing and thawing as a pretreatment method before pressing affects the intensification of obtaining juice from vegetables and fruits, such as carrots, rhubarbs, and apples. Piłat and Zadernowski [40] and Trappey et al. [41] also observed an increase in the pressing efficiency in the case of thawed fruit. In the process of pulp pressing (procedures B and C), the pressing yield increased by approximately 68%, whereas in the case of pressing chips, the yield increased three times in both procedures B and C in comparison to procedure A. In procedures B and C with samples in the form of chips and pulp, the total yield amounted to approximately 75%. A slightly higher value (78.2%) during the pressing process of carrot roots with the use of a hydraulic press in laboratory conditions was found by Sharma et al. [38]. This value is higher in comparison with the efficiency of pressing for root vegetables without enzymatic treatment in industrial conditions. For instance, when producing carrot juice, the pressing yield amounts to 50–65% [40].

Analysis of the first pressing stage proved that the pressing yield $Y_1$ of the pulp without freezing and thawing (procedures A and B) in relation to the yield of chip pressing is 87.5% higher. After a pretreatment procedure consisting of freezing and thawing the raw material before pressing, the yield of the pulp pressing increased by approximately 37.8% (procedure C), and in the case of chips pressing, it increased nearly 2.5 times (Figure 4). There were no statistically significant differences between the pulp and chips pressing yield in procedure C.

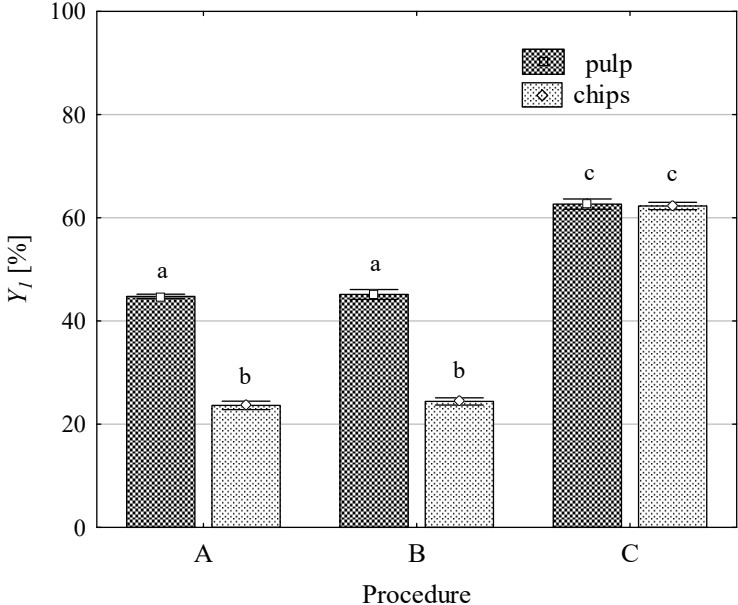

**Figure 4.** Impact of applied procedures on yield $Y_1$ in the first pressing cycle of ground celeriac root. Abbreviations, as in Figure 3. a, b, and c—average values marked with the same letter show no statistically significant difference ($p \leq 0.05$).

In the second pressing stage, the highest yield $Y_2$ was recorded using procedure B. In this case, 20.8% higher pressing yield was recorded for chips compared to the pulp (Figure 5).

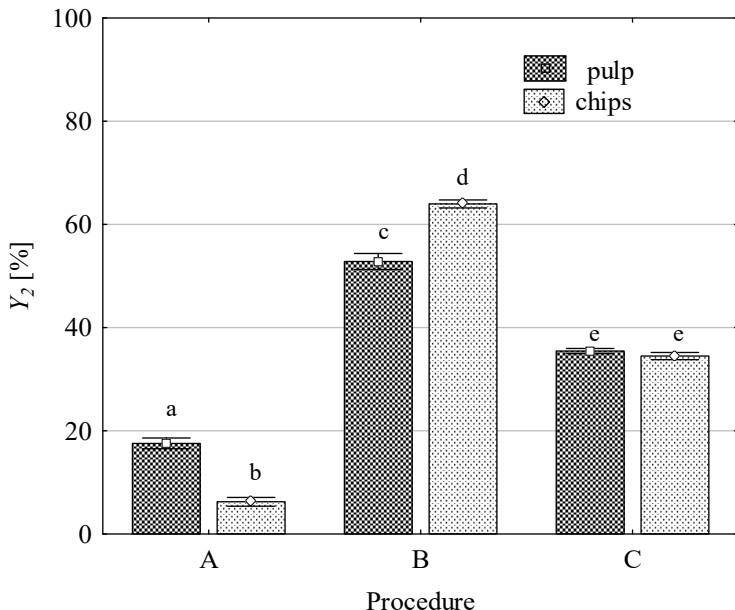

**Figure 5.** Impact of applied procedures on yield $Y_2$ in the second pressing cycle of pomace. Abbreviations, as in Figure 3. a, b, c, d, and e—average values marked with the same letter show no statistically significant difference ($p \leq 0.05$).

One of the significant parameters characterizing pressing is its energy consumption, which is presented in the form of specific energy. We only analyzed the specific energy of pressing in our study. The specific energy of celeriac root grinding, as well as its freezing and thawing, were not included. Figure 6 presents the influence of procedures used in the experiment on the specific energy $E_t$ of the pressing juice from celeriac pulp and chips. Differences between the values of specific energy $E_t$ occurring in individual procedures in the case of pulp pressing were statistically significant. However, no differences between the specific energy during pressing chips in accordance with procedures B and C were found. Energy consumption in the case of unfrozen celeriac root chips was 4.3 times higher than in unfrozen pulp (procedure A). The lowest value of energy consumption was observed in the process of pulp pressing in accordance with procedure B.

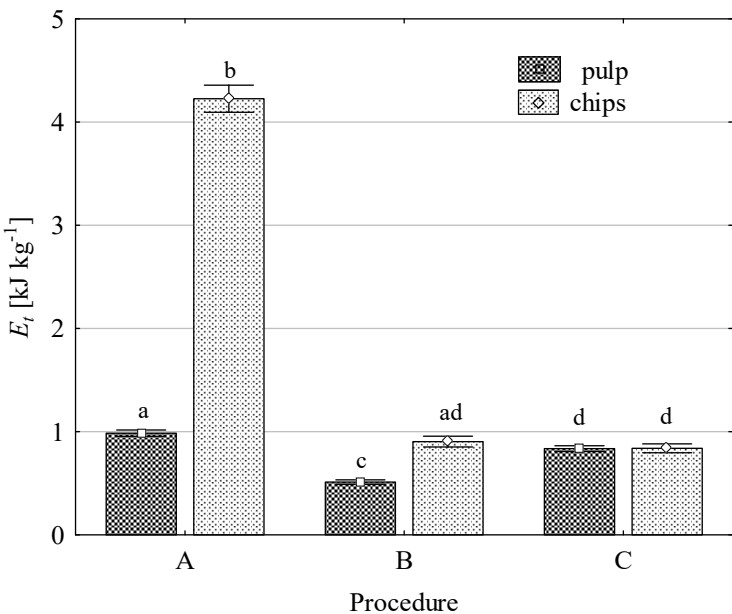

**Figure 6.** Impact of applied procedures on the total specific energy $E_t$ of pressing juice from celeriac. Abbreviations, as in Figure 3. a, b, c, and d—average values marked with the same letter show no statistically significant difference ($p \leq 0.05$).

Our analysis of the first pressing stage proved that freezing and thawing pulp or chips as a pretreatment before pressing does not significantly affect the specific energy (Figure 7). A major impact of the raw material grinding method and the selected procedure on specific energy $E_2$ was observed during the second pressing cycle (Figure 8).

The analysis proved that the yield of pressing the ground root into pulp and chips after applying procedures B and C did not show statistically significant differences. This proves the possibility of using a lower grinding degree before pressing with the use of freezing and thawing processes. Applying procedure B is advisable while comparing energy consumption and complexity of procedures B and C. Celeriac roots are a problematic material to grind due to their specific structure. It requires more specific energy during the grinding process in comparison to other root vegetables. The energy consumption of root vegetable pressing is determined by their grinding degree [38]. Our study proves that with the use of the freezing and thawing method, the degree of raw material grinding is not the most significant factor in energy consumption decrease (Figure 6) and pressing yield increase (Figure 3).

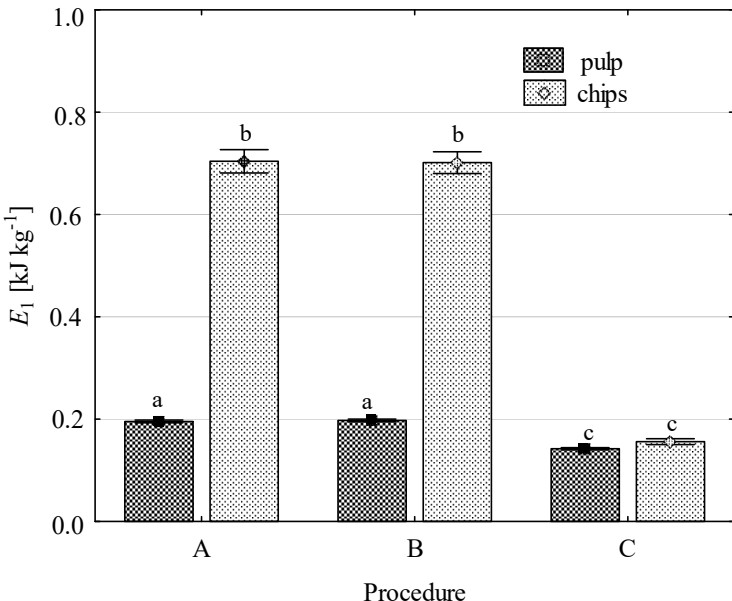

**Figure 7.** Impact of applied procedures on the specific energy $E_1$ in the first pressing cycle of ground celeriac root. Abbreviations, as in Figure 3. a, b, and c—average values marked with the same letter show no statistically significant difference ($p \leq 0.05$).

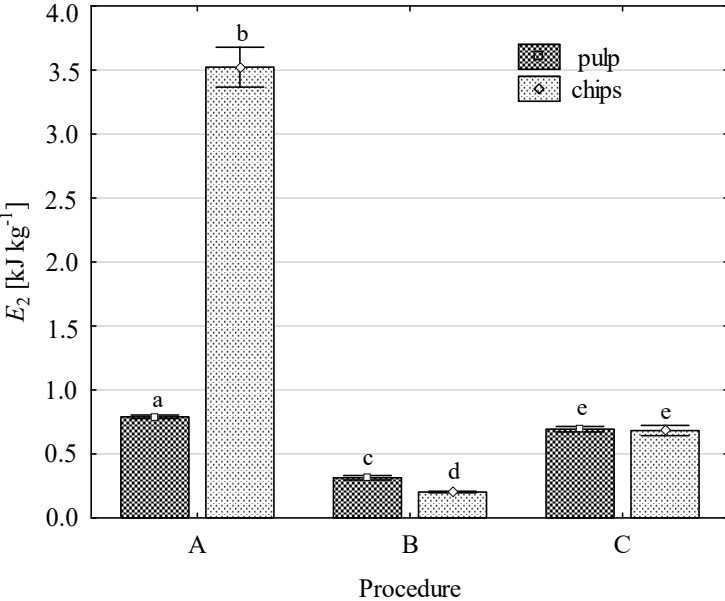

**Figure 8.** Impact of applied procedures on the specific energy $E_2$ in the second pressing cycle of pomace. Abbreviations, as in Figure 3. a, b, c, d, and e—average values marked with the same letter show no statistically significant difference ($p \leq 0.05$).

The use of freezing and thawing as a pretreatment before the pressing process causes the destruction of cells and, as a result, increases the efficiency of pressing and reduces the energy consumption of the process. The strength properties of plant materials subsequently to thawing depend on the freezing rate [42–44]. In our experiment, slow freezing was used to destroy cellular structures. Generally, water that crystallizes in vacuoles during freezing results in the rending of thin cytoplasmic membranes that surround vacuoles as well as the cell membrane. During the slow freezing process, sizable ice crystals form between spaces in the cell membrane as a result of cell juice entering the spaces [45,46]. Leakage of juice from cells results in tissue weakness and loss of stiffness. Large crystals forming during the slow freezing process have sharp edges which damage cellular components (e.g., organelles,

cell membrane, etc.) due to the pressure occurring while the crystals are growing [47,48]. As a result of thawing, the cells lose turgor pressure and become delicate, which causes cell juice leakage. During the pressing process, after sample thawing, further damage of already weak cells and tissues of celeriac root occur, which results in the release of more cell juice. A violation of the continuity of structures reduces flow resistance and thus facilitates the extraction of juice.

*3.2. Juice Properties*

A statistical analysis proved the impact of the material grinding method and the selected procedure on the soluble solids content and juice pH. Higher content of juice extract was obtained from pulp compared to chips, regardless of which procedure was used. The soluble solids content in the juices obtained as a result of pressing the pulp from celery root twice in accordance with procedures A and B did not show statistically significant differences, whereas, in the case of juice obtained in accordance with procedure C, the juice soluble solid content was approximately 6% lower (Table 1). The soluble solids content in juices pressed from celeriac root in accordance with procedures A and B was lower by approximately 17.4% and 20.4% in comparison with the value of the extracted content obtained in accordance with procedure C (Table 2). Similar results of the study concerning carrot juice were obtained by Nadulski et al. [13]. Juice obtained from fresh carrot roots had a soluble solid content that was a few percents lower compared to juice obtained from chips with the use of freezing and thawing as a pretreatment method before pressing.

**Table 1.** Extract content (°Bx) of juice from celeriac root pulp according to the selected procedure and pressing stage. Abbreviations, as in Figure 3.

| Stage | Procedure A | Procedure B | Procedure C |
|---|---|---|---|
| 1st pressing | Juice JA1 8.93 ± 0.16 a | Juice JB1 8.97 ± 0.19 a | Juice JC1 8.34 ± 0.10 b |
| 2nd pressing | Juice JA2 9.14 ± 0.21 a | Juice JB2 9.01 ± 0.09 ab | Juice JC2 8.86 ± 0.19 b |
| Total | Juice JA 8.98 ± 0.16 a | Juice JB 8.98 ± 0.15 a | Juice JC 8.43 ± 0.10 b |

a, b—mean values in the rows marked with the same letter do not have statistically significant differences ($p \leq 0.05$).

**Table 2.** Extract content (°Bx) of juice from celeriac root chips according to the selected procedure and pressing stage. Abbreviations, as in Figure 3.

| Stage | Procedure A | Procedure B | Procedure C |
|---|---|---|---|
| 1st pressing | Juice JA1 6.36 ± 0.10 a | Juice JB1 6.42 ± 0.06 a | Juice JC1 7.89 ± 0.14 b |
| 2nd pressing | Juice JA2 6.35 ± 0.13 a | Juice JB2 6.69 ± 0.11 b | Juice JC2 8.50 ± 0.14 c |
| Total | Juice JA 6.36 ± 0.09 a | Juice JB 6.60 ± 0.09 b | Juice JC 7.99 ± 0.13 c |

a, b—mean values in the rows marked with the same letter do not have statistically significant differences ($p \leq 0.05$).

The juice obtained from celeriac roots in laboratory conditions had a similar pH value (Table 3), which was, however, lower by 10–12% in the study concerning celeriac root conducted by Novotna et al. [9]. In the case of juice obtained from chips with the use of the freezing and thawing method, the pH value was slightly lower (Table 4).

**Table 3.** pH of juice from celeriac root pulp according to the selected procedure and pressing stage. Abbreviations as on Figure 3.

| Stage | Procedure A | Procedure B | Procedure C |
|---|---|---|---|
| 1st pressing | Juice JA1<br>5.65 ± 0.06 a | Juice JB1<br>5.64 ± 0.05 a | Juice JC1<br>5.89 ± 0.06 b |
| 2nd pressing | Juice JA2<br>5.67 ± 0.05 a | Juice JB2<br>5.52 ± 0.05 b | Juice JC2<br>5.47 ± 0.05 b |
| Total | Juice JA<br>5.66 ± 0.6 a | Juice JB<br>5.58 ± 0.05 b | Juice JC<br>5.73 ± 0.05 c |

a, b, c—mean values in the rows marked with the same letter do not have statistically significant differences ($p \leq 0.05$).

**Table 4.** pH of juice from celeriac root chips according to the selected procedure and pressing stage. Abbreviations as on Figure 3.

| Stage | Procedure A | Procedure B | Procedure C |
|---|---|---|---|
| 1st pressing | Juice JA1<br>6.06 ± 0.07 a | Juice JB1<br>6.03 ± 0.07 a | Juice JC1<br>6.01 ± 0.06 a |
| 2nd pressing | JA2<br>6.05 ± 0.06 a | JB2<br>5.80 ± 0.06 b | JC2<br>5.55 ± 0.05 c |
| Total | Juice JA<br>6.06 ± 0.06 a | Juice JB<br>5.85 ± 0.06 b | Juice JC<br>5.84 ± 0.05 b |

a, b, c—mean values in the rows marked with the same letter do not have statistically significant differences ($p \leq 0.05$).

Generally, juices obtained from the pulp in comparison with juices obtained from chips have a higher extract content and slightly lower pH, which is valuable from a technological perspective.

## 4. Conclusions

Among the examined methods of obtaining juice, the most beneficial method was pressing juice from the pulp, then freezing and thawing the pomace obtained in the first cycle, and finally, pressing the pomace (procedure B). This is an energy-optimal method that guarantees a high pressing yield as well as obtaining juice with a high extract content—higher than during the process of pressing chips. Certainly, this method has additional costs associated with freezing and thawing, but there is no chemical impact on the raw material as occurs in the case of enzymatic treatment. Our study indicates the possible use of freezing and thawing methods as alternative techniques for obtaining juice from vegetables with pro-health properties. Analyzed methods of cold juice pressing from celeriac root without the use of enzymatic treatment guarantee that obtained juices will maintain their pro-health properties. Further research on the use of freezing and thawing in the process of vegetable juice pressing is essential, as well as the assessment of nutritional and organoleptic properties of obtained juice.

**Author Contributions:** Conceptualization, R.N. and Z.K.; methodology, R.N.; formal analysis, R.N. and Z.K.; investigation, T.G., R.N., and Z.K.; data curation, T.G. and Z.K.; writing—original draft preparation, R.N., Z.K., and T.G.; writing—review and editing, R.N. and Z.K.; supervision R.N. All authors have read and agreed to the published version of the manuscript.

**Funding:** This research received no external funding.

**Conflicts of Interest:** The authors declare no conflict of interest.

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
