# Peer review of "The Influence of Freezing and Thawing on the Yield and Energy Consumption of the Celeriac Juice Pressing Process"

_processes, doi:10.3390/pr8030378_

Round 1

Reviewer 1 Report

In the presents study authors have studied the use of freezing and thawing as a method of supporting the process of pressing celeriac roots. The main target of the studied focused on the influence of pretreatments consisting of grinding vegetables and whether freezing and thawing of the raw material would affect the quality of the obtained juice. It is an interesting study thought it was manufacture by lab equipment. This kind of studies would provide more important outcomes if they are conducted in regard with the industrial parameters. Authors are advised to compare their parameters with ones utilized by the industry. In addition, some minor point need to be addressed prior publication:

Minor points

  1. What to authors mean by “specific energy” on the title of the manuscript? Please revise as it does not provide the content of the manuscript clearly
  2. Line 10: Please avoid statements like “In the literature there are no publications”. You can use statements as “to the best of our knowledge the are no evidence regarding…” Authors are advised to use such statements in the introduction or in the discussion section not in the abstract.
  3. Line 13: Add the missing dot. Check this manner though the whole manuscript
  4. Line 22: Please avoid “:” within the abstract section. Revise the sentence accordingly
  5. Authors are advised to provide a graphical abstract showing the procedure and final product.
  6. Line 75: authors are advised to provide the specific info regarding the area they collected the celery roots (which farm; which place etc). Also, authors can provide some ingredient information regarding the roots the utilized. For example, humidity would be a crucial factor.
  7. Lines 78-85: Was this method based on previously published data. Please add the appropriate references.
  8. Lines 129-133: Was the procedure tested in duplicate or triplicate?
  9. Line 145: Please provide an explanation regarding A, B and C within the Figure caption. Do that in all figures accordingly.

Reviewer 2 Report

Manuscript 745550

Title: The influence of freezing and thawing on yield and specific energy of celeriac juice pressing process

Journal: Processes

The manuscript titled “The influence of freezing and thawing on yield and specific energy of celeriac juice pressing process” describes the effect of freezing pre-treatment on juice yield and specific energy required for the pressing process. Data on the juice characteristics such as pH and solid content is also reported. The results are quite interesting and the experimental plan is well-defined. The work is acceptable for publication after minor revisions. Some issues should be clarified.

The authors determined the extract content of the juice. Is it the total solid content determination? Please verify and correct throughout the manuscript.

Why the authors did not evaluate other juice quality parameters such as phenolic compounds, bioactive compounds? The choice of one of the proposed processing scheme should be based on the overall quality of the juice obtained.

The work focused on the juice. Is there a potential application for the recovery of bioactive compounds from pomace? If yes, please describe in the manuscript.

Did the authors consider the possibility of a combined approach with the use of enzymes?

Other papers could be mentioned in the manuscript. The following papers, but not limited to, are suggested: doi.org/10.1300/J492v07n03_06; Piłat, B., & Zadernowski, R. (2016). Effect of freezing and enzymatic treatment on the output of juice and extraction of phenolic compounds in the pressing of lingonberry fruit. Pol J Nat Sci, 31(3), 413-420; doi.org/10.1080/10408398.2015.1132670
